# Tackling Sleeping Sickness: Current and Promising Therapeutics and Treatment Strategies

**DOI:** 10.3390/ijms241512529

**Published:** 2023-08-07

**Authors:** Miebaka Jamabo, Maduma Mahlalela, Adrienne L. Edkins, Aileen Boshoff

**Affiliations:** 1Biotechnology Innovation Centre, Rhodes University, Makhanda 6139, South Africa; miebakajamabo@yahoo.co.uk (M.J.); madumamahlalela@gmail.com (M.M.); 2Department of Biochemistry and Microbiology, Biomedical Biotechnology Research Centre (BioBRU), Rhodes University, Makhanda 6139, South Africa; a.edkins@ru.ac.za

**Keywords:** African trypanosomiasis, *Trypanosoma brucei*, sleeping sickness, drug discovery, drug target, COVID-19, drug resistance

## Abstract

Human African trypanosomiasis is a neglected tropical disease caused by the extracellular protozoan parasite *Trypanosoma brucei*, and targeted for eradication by 2030. The COVID-19 pandemic contributed to the lengthening of the proposed time frame for eliminating human African trypanosomiasis as control programs were interrupted. Armed with extensive antigenic variation and the depletion of the B cell population during an infectious cycle, attempts to develop a vaccine have remained unachievable. With the absence of a vaccine, control of the disease has relied heavily on intensive screening measures and the use of drugs. The chemotherapeutics previously available for disease management were plagued by issues such as toxicity, resistance, and difficulty in administration. The approval of the latest and first oral drug, fexinidazole, is a major chemotherapeutic achievement for the treatment of human African trypanosomiasis in the past few decades. Timely and accurate diagnosis is essential for effective treatment, while poor compliance and resistance remain outstanding challenges. Drug discovery is on-going, and herein we review the recent advances in anti-trypanosomal drug discovery, including novel potential drug targets. The numerous challenges associated with disease eradication will also be addressed.

## 1. Introduction

African trypanosomiasis is a neuropathic and wasting disease, affecting both humans and animals, and endemic to Sub-Saharan Africa [1,2,3]. The causative agents of the disease in humans are subspecies of the extracellular vector-borne parasite *Trypanosoma brucei,* which is a protozoan parasite transmitted through the bite of an infected tsetse fly (*Glossina* species) during a bloodmeal [2]. The human infective forms, *T. brucei gambiense* (*T. b. gambiense*) and *T. brucei rhodesiense* (*T. b. rhodesiense*), are phenotypically indistinguishable and cause human African trypanosomiasis (HAT), commonly known as sleeping sickness, which is a neglected tropical disease (NTD) prevalent in Sub-Saharan Africa. *T. b. gambiense* is anthroponotic as humans are the main reservoir, while *T. b. rhodesiense* is zoonotic since its transmission relies on animal reservoirs [4]. The chronic infection caused by *T. b. gambiense* accounts for 98% of reported cases (referred to as *G*-HAT), while the acute zoonotic infection caused by *T. b. rhodesiense* is responsible for 2% of the reported cases (referred to as *R*-HAT) [5]. *T. b. gambiense* and *T. b. rhodesiense* are the only trypanosomes in Africa that have been reported to successfully establish an infection in humans [6]. *T. brucei brucei* (*T. b. brucei*) is one of several species of trypanosomes that cause *nagana* or Animal African Trypanosomiasis (AAT) in both domestic and wild animals [7].

All species and subspecies in the genus *Glossina* have the ability to transmit trypanosomes to humans, but some groups have been particularly adapted for transmission of specific species; the *Glossina palpalis* and *Glossina fuscipes* groups for *T. b. gambiense* and the *Glossina morsitans* and *Glossina pallidipes* groups for *T. b. rhodesiense* [8,9]. Although parasite transmission is primarily due to the bite of the tsetse fly, other modes of transmission have been reported, such as mother-to-child transmission through the placenta (for *T. b. gambiense*), mechanical transmission by other hematophagous insects, mishandling of laboratory samples, and sexual intercourse involving infected individuals [10,11,12].

It is difficult to determine the exact number of cases of the disease, as most infections now occur in remote rural foci. In April 2020, the World Health Organization (WHO) postponed all active screening campaigns for neglected tropical diseases; however, this recommendation was later revised to support NTD programs in the context of the COVID-19 pandemic. The impact of delayed screening in previous years had led to a resurgence of HAT in the endemic communities. The COVID-19 pandemic contributed to the lengthening of the proposed time frame of eliminating HAT as the control programs were interrupted and funding was diverted. It is predicted that the Democratic Republic of Congo (DRC) may experience the greatest risk due to several foci with high disease prevalence, coupled with violent conflict. If the goal to eliminate the transmission of HAT by 2030 is to be achieved, then urgent strategies need to be put in place to get treatment and screening campaigns back on track, to alleviate the losses experienced during COVID-19. Other aggravating factors include climate change, the increased movement of people and animals, and the urgent need for less complicated methods of diagnosis and treatment. Despite the recent development of a new drug, fexinidazole, novel anti-trypanosomal drugs are still being explored.

## 2. Disease Burden of Human African Trypanosomiasis

HAT is endemic to the Sub-Saharan African region, with rural farming communities being the most vulnerable [13,14]. The disease is distributed across 36 Sub-Saharan African countries, with *T. b. gambiense* present in 24 countries in Central and Western Africa, while *T. b. rhodesiense* is found in 13 countries in East and Southern Africa (Figure 1) [4,13,15,16]. Only in Uganda have both *T. b. gambiense* and *T. b. rhodesiense* been reported to occur, albeit in different regions: *T. b. gambiense* in the northwest and *T. b. rhodesiense* in the central regions (Figure 1) [17,18]. Knowledge of the tsetse fly involves their specific control methods, climate involvements, and other demographics that have significant roles to play in the prevalence of sleeping sickness in these areas [19]. The geographical prevalence of trypanosomiasis in Uganda, over the years, has shifted focus to the West Nile areas of Uganda [20,21].

HAT has caused millions of fatalities due to numerous outbreaks, which have led to epidemics. Over the course of the past two centuries, there have been three major epidemics in Africa. The first epidemic occurred between 1896 and 1906, killing nearly a million people [22,23]. East African countries such as Uganda and Kenya were severely affected by what was then assumed to be *T. b. gambiense*, as *T. b. rhodesiense* was yet to be characterized [24]. The second epidemic occurred about 10 years later and resulted in continuous surveillance strategies of the population, as well as vector control measures. These strategies were effective enough to almost eradicate the disease in less than 50 years [25]. However, a decline in surveillance measures and civil unrest in these countries led to a resurgence of the disease in numerous countries in Central Africa [26,27,28,29]. After the peak in the early 21st century, sustained efforts by agencies such as WHO to control *G*-HAT again included providing free drugs to affected areas, supporting and strengthening vector/disease control measures, and improving knowledge of the disease [30]. Consequently, the reported cases have dropped considerably over the years, from 40,000 cases in 1998 to 9878 cases in 2009. In 2019, the number of reported cases declined to 992, with over 70% of cases reported between 2009 and 2019 having been recorded in the DRC [31]. Despite the strides made in combating the disease, approximately 60 million people are still at risk of contracting HAT [2,32].

Despite the declining numbers of reported cases in the past decades, WHO indicates a wide discrepancy between the actual numbers of cases compared to the reported cases, due to poor coverage from inefficient surveillance systems [33]. There are reports of “blind spots”, where no HAT control activities have taken place in the past few decades in very remote areas of Central African Republic and the DRC. With areas such as these accounting for “invisible” cases, HAT elimination is complicated further. Incidental cases of HAT do get reported in non-endemic territories, even outside of Africa, which are attributed to infected individuals travelling from endemic countries [34]. Climate change is also expected to play an important role in the epidemiology of HAT, as global temperatures continue to rise [35].

## 3. Lifecycle of *Trypanosoma brucei*

*T. brucei* cycles between two obligatory hosts, the tsetse fly and the mammalian host, to complete its life cycle. Both the male and female blood-feeding tsetse fly can cause transmission [25]. The parasite morphotypes in the human and tsetse fly are referred to as the bloodstream (BSF) and procyclic (PCF) forms, respectively. The tsetse fly takes up the bloodstream trypomastigotes from the blood of the mammalian host while feeding, and the trypanosome multiples in the midgut, producing replicative procyclic trypomastigotes that enable the survival of the trypanosome in this environment (Figure 2) [36]. After complex adaptations through the tissues of the tsetse fly, parasites leave the midgut as epimastigotes and travel to the salivary glands where they multiply and form the short and stumpy metacyclic trypomastigotes, which are human infective. The parasite is then transmitted to a mammalian host from the bite of an infected fly (Figure 2) [37,38]. The BSFs of the parasite are either long slender (LS) or short stumpy (SS). The LS bloodstream forms are proliferative, adapted for optimal tissue invasion, and possessing the ability to traverse blood vessels and enter perivascular spaces [39,40]. During infection within the mammalian host, the LS BSFs differentiate into the quiescent SS BSFs. This is to ensure that the mammalian host lives long enough so that the parasites can be transmitted to another vector, which will in turn spread them to other mammalian hosts [4,41]. The transition from the LS morphology to the SS bloodstream forms is driven by a quorum-sensing pathway whereby high population densities of the LS BSFs result in the release of the stumpy-inducing factor [42]. The differentiation to the LS morphology is to ensure parasite survival, as the SS morphology is not optimally adapted for survival in the mammalian host for extended periods of time [43].

Generally, in a population of tsetse flies, the population carrying the infectious metacyclic form in their salivary glands is less than 0.1%, but with its feeding pattern within a 2–3 month lifespan, it can infect many people [38]. In the mammalian host, after the tsetse fly injects the metacyclic trypomastigote form before its bloodmeal, the trypanosomes first proliferate in the tissues at the site of the infection [39]. The parasites are transmitted as the tsetse fly injects its saliva in order to inhibit blood clotting and vasoconstriction [40,44,45,46].

One of the morphological markers is the position of the flagellar pocket relative to the nucleus [47,48]. *T. brucei* is restricted to the trypomastigote and epimastigote morphotypes, with the flagellar pocket of the trypomastigotes positioned at the posterior end of the cell opposed to the center in epimastigotes [2]. These morphotypes of *T. brucei* are characterized by having a laterally fixed flagellum [49].

*T. brucei* is an extracellular parasite generally shown to reside and multiply in the interstitial spaces, lymphatic system, and bloodstream of tissues of the mammalian host [4,50]. However, several other tissues have been found to serve as reservoirs for the *T. brucei* parasite, including the skin and adipose tissues [51,52,53]. An adipocyte tissue form, similar in morphology to the BSF, has been reported that invades mammalian fat tissues, and can utilize exogenous fatty acids, such as myristic acid, as a carbon source, and this may be a causative factor in the weight loss (wasting) that is observed in HAT sufferers [47]. Within the adipose tissues, the parasites occupy interstitial spaces, either between adjacent adipocytes or flanked by an adipocyte and a capillary [47]. The parasites possibly scavenge free fatty acids released by the adipocytes via an unknown mechanism [54]. *T. brucei* can carry out β-oxidation of fatty acids, but it is unknown if this pathway serves to fulfill the energy requirements of the adipose tissue forms [47]. A subpopulation of metacyclic trypomastigotes is also retained intradermally, within the vicinity of the bite, which remains highly infectious, even in the absence of detectable parasites in the blood [51]. In addition to the skin and adipose tissues, there is also evidence of the parasite residing in the testes, leading to sexual transmission, as shown in mice [55,56]. These reservoirs could be an extra layer of protection of the parasite from the host system, and a reason for continuous relapse in infected individuals. At the advanced stage of the parasitic infection, the parasites are also present in the cerebrospinal fluid, having infiltrated the central nervous system (CNS) [25].

As the parasite shuttles between the parasite vector and its mammalian host, it undergoes changes in its gene expression to provide proteins that are adapted to function in each host [57]. The BSFs uses glucose through the glycolytic pathway in the glycosome as they can barely survive in anoxic conditions, whereas the PCFs makes use of amino acids, such as proline and threonine, as carbon source through the Krebs cycle in the mitochondrion [58].

## 4. Symptoms and Disease Progression

The infected patient exhibits few symptoms immediately after being infected, but symptoms develop as the parasite multiplies in the blood and lymphatic vessels. The symptoms of HAT vary according to the stage of the parasitic infection (Figure 3) [2,59]. After the parasite transmission into the human bloodstream, headaches, fatigue, general malaise, and fever occur [13]. At the site of the infection, a lesion known as a trypanosomal chancre may also occur, particularly in cases of *R*-HAT [60]. As the infection progresses, the parasites invade lymph nodes, resulting in lymphadenopathy (Figure 3) [14]. The associated symptoms are heightened fever, chills, and hepatosplenomegaly [14,61]. At the phase of bloodstream and lymph node infection, the disease is said to be at the hemolymphatic stage [25]. In an advanced infection, the parasites breach the blood–brain barrier (BBB) to infiltrate the CNS, producing the meningoencephalitic stage (Figure 3) [25], which is characterized by mental, neurological, and sensory degeneration. The symptoms include, but are not limited to, delirium, disorientation, apathy, anxiety, emotional instability, abnormal speech, paresthesia, anesthesia, convulsions, seizures, and coma [13,61]. The meningoencephalitic phase of the disease is also symptomized by a disruption of the circadian cycle, resulting in daytime somnolence and nocturnal insomnia (Figure 3), hence the name sleeping sickness. If not treated, the meningoencephalitic stage of HAT can lead to coma and eventual death [4,59].

An early meningoencephalitic (“intermediate”) stage of the disease has also been suggested, whereby the parasitic infection has breached the BBB but has not yet infiltrated the brain parenchyma [62,63,64]. The rate of disease progression differs between *R*-HAT and *G*-HAT [32]. For *R*-HAT, the hemolymphatic stage symptoms appear 1 to 3 weeks following the bite of the tsetse fly, while the symptoms manifest much later in the case of *G*-HAT [13,60]. The onset of the meningoencephalitic stage of the disease occurs within 2–60 days of infection in cases of *R*-HAT, and death can take place within 3 months [14]. For *G*-HAT, the onset of the meningoencephalitic stage occurs between 300 and 500 days from infection, and the disease may persist for years [25,65].

In addition to the symptoms, especially in cases of *R*-HAT, cardiac and endocrine issues such as myocarditis and hypogonadism may also occur [66,67,68].

**Figure 3 ijms-24-12529-f003:**
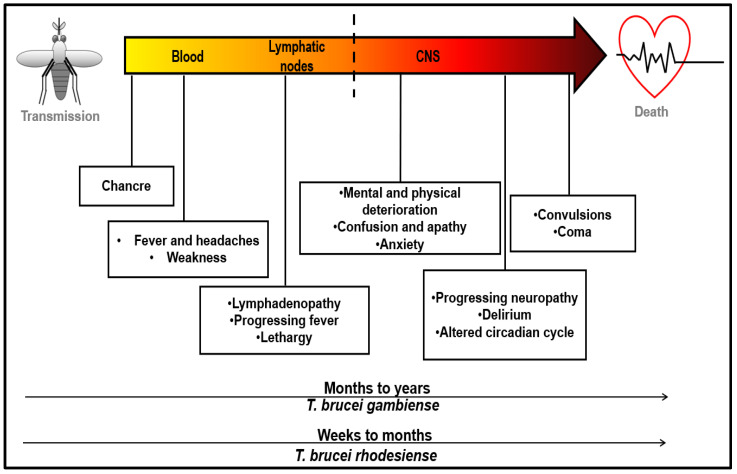
Progression of human African trypanosomiasis. The diagram highlights some of the defining symptoms of HAT as they relate to the progression of the *T. brucei* infection. The dashed line demarcates the point at which the parasitic infection infiltrates the CNS. The arrows at the bottom of the infographic indicate the different rates at which *R*-HAT and *G*-HAT progress. Adapted from [69].

## 5. Disease Diagnosis

Timely and accurate diagnosis is of paramount importance. Early diagnosis is key to ensuring that the patient receives treatment before the onset of the meningoencephalitic stage of the disease, while accuracy prevents a misdiagnosis, since some symptoms mirror those of malaria [64]. To prevent misdiagnosis, a microscopic analysis is carried out on the buffy coat of the blood sample [39]. Prior to microscopic analysis, the parasites may also be eluted from the blood sample by means of the mini anion exchange centrifugation technique (mAECT), in cases of low trypanosome concentrations [39,70,71]. The above-mentioned diagnostic techniques are not always useful for *G*-HAT diagnosis. Therefore, the card-agglutination trypanosomiasis test (CATT) is also carried out [72]. This diagnostic technique is specific to *T. b. gambiense*, detecting the presence of antibodies to VSG variants LiTat 1.3 and LiTat 1.5 [64,73]. Since treatment for HAT is infection-stage-specific, lumbar punctures are also administered as a supplementary technique, to determine if the infection has infiltrated the CNS [64]. In cases of low trypanosome concentrations in the blood, diagnosis may also be carried out using lymphatic fluids drawn from the swollen cervical lymph nodes of patients [39].

## 6. Anti-Trypanosomal Drug Treatments and Resistance Mechanisms

Even though African trypanosomes were first described more than a century ago, only a few efficacious drugs have been approved for treatment. These drugs present challenges that include toxicity, difficult administration, and trypanosome resistance. Administering the drugs is costly and labor intensive in some cases, a disadvantage considering that HAT is endemic to remote and resource-lacking regions [5,59]. At present, there is no umbrella treatment drug or strategy for HAT. The anti-trypanosomal chemotherapeutic treatment that is administered needs to be specific to whether the disease is at the hemolymphatic or meningoencaphalitic stage and whether the patient is infected with *T. b. gambiense* or *T. b. rhodesiense* [2,5,25].

### 6.1. Approved Drug Treatments

The approved drugs used to treat HAT have differing modes of action. The hemolymphatic stage of the disease is treated with pentamidine isethionate (pentamidine) and suramin for *G*-HAT and *R*-HAT, respectively. The meningoencaphalitic stage is treated with melarsoprol (Mel B or Arsobal^®^) and eflornithine (alpha-difluoromethylornithine, abbreviated to DFMO). DFMO is used in cases of meningoencephalitic *G*-HAT, while melarsoprol is used in cases of both *G*-HAT and *R*-HAT [2,5]. To expedite treatment of meningoencephalitic *G*-HAT, DFMO is also being used as a combinatorial drug in conjunction with nifurtimox (Lampit^®^) in what has been dubbed the nifurtimox–eflornithine combinatorial therapy (NECT). Nifurtimox is an independent drug that has not been authorized for the treatment of HAT, but is rather used for treating Chagas disease, that is caused by *Trypanosoma cruzi*, an American trypanosome [74,75,76,77].

The distinguishing factor between haemolymphatic and meningoencephalitic HAT treatment is whether the drugs are able to cross the BBB [78,79]. Pentamidine, which is indicated for hemolymphatic HAT, is BBB penetrative [80]. Pentamidine, which is administered through intramuscular injections, is an aromatic diamine, and its anti-trypanosomal activity dates back to the 1930s. Having been reported as effective against early meningoencephalitic HAT (“intermediate HAT”), the drug actively diffuses into the CNS in a process actively facilitated by the mammalian organic cation transporter 1 (OCT1) at the BBB [81,82]. This drug hinders DNA replication by influencing topological changes in the parasite’s DNA, thereby inhibiting normal topoisomerase functioning [2]. Pentamidine accumulation in trypanosomal cells also hinders mitochondrial activities [2,5]. The inadequacy of pentamidine as a trypanocide in the CNS is attributed to the drug accumulating at the capillary endothelium, and its active ejection back into the bloodstream by mammalian BBB ABC transporters, such as P-glycoprotein and multidrug resistance-associated proteins [80,82].

Suramin, dating back to the 1920s, is a polysulfonated naphthalene dye that inhibits glycosome-based glycolysis by interacting with selected enzymes that include 6-phosphogluconate dehydrogenase. Suramin anti-trypanocidal activity is also thought to be brought about by inhibiting low-density lipoprotein uptake, consequently having an adverse effect on the parasite’s cholesterol and phospholipid supply. Suramin is administered by means of intravenous injections. Even though suramin is only administered for treating *R*-HAT, it is also effective against *G*-HAT [2,5].

Melarsoprol, a derivative of melamine arsenical melarsen, has been used as a trypanocidal agent since the 1940s. The mode of action is not clearly understood, but it is hypothesized that the drug adversely modulates the parasite’s glycolytic and redox metabolism pathways. This drug is administered by means of an intravenous injection [5,83,84].

DFMO is a trypanocidal agent that was first used as a potential anti-cancer agent. Its use dates to the 1980s, and it acts by inhibiting ornithine decarboxylase (ODC) [85,86]. ODC is required for synthesizing polyamines that are essential for trypanosomal proliferation by cell division [86]. ODC is also necessary to produce molecules that serve as precursors of trypanothione, which is in turn necessary for the redox homeostasis of the parasite [87,88]. DFMO is administered through intravenous infusions [2,5,89]. The NECT strategy has been the main treatment of *G*-HAT since 2010. In addition to DFMO’s mode of action, nifurtimox places the trypanosomal cells under oxidative stress. This combinatorial therapy is administered orally and intravenously for nifurtimox and DFMO, respectively [5,90].

### 6.2. Drug Resistance Mechanisms

Several genotypic and molecular discoveries have been linked to observed drug resistance trypanosome phenotypes. Transporters are important for essential nutrient uptake in the *T. brucei* as parasites lack many anabolic pathways [91], such as the purine synthesis pathway, which is absent in protozoan parasites [92]. Some amino acids must be retrieved from outside of the parasites [93]. The vast majority of transporters are non-essential for parasitic survival due to redundancy, as a single substrate may enter through numerous transporters [91]. Since drug resistance is associated with transporter loss or mutation, it is hypothesized that the essential nutrients may still enter through other closely related or dissimilar transporters, which are non-permissive for the drugs [91]. Transporters that are responsible for drug resistance or treatment failure are non-essential for parasite survival; thus, receptor-mediated endocytosis is viewed as a drug target that could be manipulated for drug delivery [94].

The resistance of trypanosomes to pentamidine and melarsoprol is linked to a phenomenon referred to as melarsoprol/pentamidine cross resistance (MPXR) [95]. The implicated entities in MPXR are the aminopurine transporter P2 (encoded by *TbAT1*), the low-affinity pentamidine transporter (LAPT1), and the high-affinity pentamidine transporter (HAPT1; aquaglyceroporin2) [95]. With regard to the P2 transporter, it has been determined that its deletion or loss only partially influences MPXR, whereby HAPT1 loss is the primary determinant for high-level MPXR [95,96,97,98]. LAPT1 has a significantly lower affinity for pentamidine compared to the P2 transporter and HAPT1 [99,100]. HAPT1 has the highest affinity for pentamidine, while the P2 transporter is the primary factor in melarsoprol uptake [99,100].

There are two orthologues of HAPT1 in *T. brucei*, aquaglyceroporins 1 and 3 (AQP1 and AQP3) [98,101]. Field studies have shown that *T. brucei* mutants exhibiting AQP2 and AQP3 chimerization are responsible for pentamidine and melarosprol resistance, even in the absence of mutations or losses of the P2 transporter [102,103,104]. AQP3 is inconsequential in terms of MPXR [96]. AQP3 is localized at the cell body membrane and AQP1 at the flagellar membrane [96,101]. AQP2 localizes at the flagellar pocket and flagellar membrane in the BSFs and PCFs, respectively [96]. Since endo- and exocytosis are increased at the flagellar pocket, it is possible that the localization of AQP2 could be a factor in the transporter being the main determinant of pentamidine and melarsoprol uptake. The ability of AQP2 to transport the drugs is attributed to its enlarged atypical core, which is markedly different from those of AQP1 and AQP3 [96,100,105]. There seems to be contention when it comes to the pentamidine uptake mechanism by AQP2. Whether the drug permeates through the core of AQP2 or AQP2 serves as a receptor for the endocytic uptake of the drug is unclear [105,106].

Coupling pentamidine to PEGylated chitosan particles coated with a camel heavy-chain-derived nanobody resulted in increased drug uptake, treating HAT in mice models at 100-fold lower concentrations compared to pentamidine alone [107]. This is due to the nanobody recognizing cryptic epitopes that are presented at the trypanosomal cell surface [107,108]. The pentamidine loaded onto the PEGylated chitosan particles coated with the nanobodies (NbAn33-pentamidine-chNPs) was also trypanocidal in vitro toward resistant AQP2-lacking cell lines [107]. NbAn33-pnetamidine-chNP was also able to treat mice infected with an AQP2-lacking trypanosomal cell line [107]. AQP2-lacking trypanosomes are highly sensitive to salicylhydroxamic acid (SHAM), octyl gallate, and propyl gallate, which are all inhibitors of the trypanosome alternative oxidase (TAO) [109]. This is due to AQP2 being responsible for glycerol uptake and efflux; therefore, inhibiting TAO leads to pathways that result in a toxic accumulation of glycerol [91,109].

Surface glycoproteins have been implicated in suramin resistance. Suramin interacts with invariant surface glycoprotein 75 (ISG75), where the abundance of ISG75 is a function of suramin accumulation in the trypanosomal cells [110]. Cells displaying a particular variant surface glycoprotein (VSG), denoted as VSG^sur^, exhibited an increased resistance toward suramin [111]. A high-affinity complex is formed between suramin and VSG^sur^, and the drug is bound by the large pocket of the VSG homodimer [112]. Mutations within the suramin-binding pocket of VSG^sur^ perturb the high-affinity VSG^sur^–suramin complex [112]. The mechanisms of suramin resistance by VSG^sur^ is that the VSG impairs particular receptor-mediated endocytic pathways that are used by trypanosomes to internalize suramin [113].

VSG^sur^ is a member of a group of VSGs that exhibit unique structural properties [112], and within this group, VSG13 has been identified and structurally defined [112]. It has not been determined if VSG13 and other VSGs that fall within this structural grouping also possess suramin-binding capabilities or if they serve as determinants for suramin resistance when presented at the trypanosomal cell surface. A study of trypanosomes obtained from suramin responsive and non-responsive HAT patients attempted to elucidate the underlying proteomic profiles of resistant trypanosomes [114]. The trypanosomes isolated from the non-responsive patients (*T. b. rhodesiense*, strain EATRO-734) exhibited upregulated metabolic and detoxification processes [114]. Furthermore, the mitochondrial proteome of the EATRO-734 strain was also demonstrated to be upregulated. Inadequacies in the endosomal pathway are also implicated in suramin resistance [114]. Regarding DFMO, a single gene (*TbAAT6*) has been determined to encode the transporter of the drug, whereby the loss or knockdown of the gene results in drug resistance [115]. It is suggested that DFMO resistance may easily be detected in the field by carrying out PCR [115].

### 6.3. Physiological Challenges to Effective Drug Delivery

An increased understanding of trypanosomal biology is required to facilitate effective drug delivery as the parasite resides in different tissue types in the body, effectively serving as a long-term reservoir. Due to renewed interest in tissue-resident parasite populations, the implications of these parasite populations for future studies were reviewed [116]. During the course of infection, subpopulations of the parasites may exhibit divergent metabolic profiles where some subpopulations are quiescent, with downregulated metabolic pathways that are targeted by drugs [117]. Some subpopulations could be at different stages of differentiation; therefore, drugs targeting cell division may be ineffective. These subpopulations, which may be less susceptible to being targeted by drugs, are referred to as persisters [117,118], and may cause relapses in infectious diseases, even after treatment courses have been duly completed [118,119].

In the protozoan parasites *Plasmodium vivax* and *Toxoplasma gondii*, persister subpopulations, represented by the hypnozoite and bradyzoite morphotypes, respectively, have been reported [118,120]. In Chagas disease, the persister phenomenon is implicated in the chronicity of the disease and the frequency of non-efficacy by the front-line drugs, nifurtimox and benznidazole [121,122]. Eliminating the parasitic population in its entirety is essential for curing Chagas disease, as the persistence of low populations of the parasite could worsen the outcomes for patients by eventually resulting in sickness in asymptomatic individuals [117].

Tissue distribution also needs to be considered as African trypanosomes are dispersed across several tissues, with some tissues potentially offering protection from the immune system [123]. This is because immune-privileged and drug-impenetrable tissues may be sources of infection re-emergence, even in cases where the anti-parasitic treatment has been administered to the full term [123]. The *T. brucei* parasite in the early stage of the infection invades various sites, which apart from the bloodstream and lymph nodes, are the testes, skin, and adipose tissue [47,52,55]. These sites are believed to be immune privileged, possibly providing a safe haven for parasitic cells [123]. However, there are emerging reports of trypanosomes accumulating in adipose tissue, whereby they trigger an immune response [124]. The immune response leads to adipocyte lysis (adipolysis) [124]. In this instance, adipolysis is hypothesized to serve as a parasitemia reduction mechanism, also prolonging the lifespan of the mammalian host [47,125]. The adipose tissue forms of the parasite are quiescent, exhibiting downregulated protein synthesis, and they are less susceptible to drugs [124]. They exhibit reduced virulence and eventually become undetectable, being a possible source of disease re-emergence. The adipose tissue forms are therefore a potential source of persistence and chronicity [124]. A study found that persister quiescent skin tissue forms of the *T. brucei* parasite, which are hypothesized to be responsible for prolonged and persistent infections, were quickly established when cultured in artificial human skin, and these quiescent forms could be metabolically reactivated [126]. Therefore, the drug impenetrability of adipose tissues presents an additional challenge [47,123]. The adipose-tissue-occupying parasites are possibly more densely populated compared to those in the bloodstream and CNS [126]. Adipose- and skin-tissue-occupying parasites could be targeted via the fatty acid uptake machinery of the parasite, as it is viewed as a vehicle for drug delivery [54]. This entails coupling drugs to fatty acids, which would be lipophilic and thus optimized for skin or even adipocyte tissue penetration [54].

### 6.4. Recent Advances in Drug Development

Most recently, fexinidazole has been sanctioned for the treatment of both stages of *G*-HAT, albeit indicated for non-severe meningoencephalitic HAT [127,128,129]. Fexinidazole is a pro-drug that emerged from a Drugs for Neglected Diseases initiative (DND*i*) and Swiss Tropical and Public Health Institute (Swiss TPH) collaboration, and was further developed by Sanofi. The drug is currently the only HAT treatment that is administered orally, having efficacy at both stages of the disease [129]. Fexinidazole, like nifurtimox, is a nitro pro-drug dependent on the putative ubiquinone nitroreductase for activation in the mitochondrion. It is however unclear whether the mode of trypanocidal action is mainly due to mitochondrial function inhibition or the targeting of components outside the parasite’s mitochondrion [130,131,132,133]. Fexinidazole is being trialed for the treatment of both stages of *R*-HAT and intermediate Chagas disease. The *R*-HAT clinical trials were conducted in Uganda and Malawi at the Lwala and Rumphi district hospitals, respectively [134]. Uganda and Malawi account for more than 90% of *R*-HAT incidences. The outcomes of the fexinidazole trials against *R*-HAT are currently being prepared for publication [134]. The phase II, multicenter, randomized, and placebo controlled proof-of-concept trial of fexinidazole against intermediate Chagas has recently been published [135]. Fexinidazole was determined to effectively clear trypanosomes in Chagas patients; however, cases of neutropenia and upregulation of hepatic enzymes were also detected [135].

In addition to fexinidazole, which has been approved for clinical use, several drugs with the potential to yield the effective treatment of HAT have recently progressed to clinical trials [136]. A class of compounds, referred to as benzoxaboroles, has emerged as promising novel drugs for the treatment of HAT [137]. The compounds are derivatives of an oxaborole heterocycle (boron heterocyclic) that is coupled to a phenyl group [138,139]. The most notable benzoxaborole is acoziborole (AN5568, SCYX-7158), which is derived from gem-dimethyl 4-fluoro-2-trimethylfluoro benzamide that has progressed to phase III clinical trials for *G*-HAT, and it is leading in the pipeline of novel HAT drugs [136,140]. Orally dosed acoziborole can efficaciously and safely treat the hemolymphatic and meningoencephalitic stages of *G*- and *R*-HAT [141]. SCYX-7158 is metabolized into SCYX-3109 by trypanosomatids, with its cellular target being the nuclear-localized cleavage and polyadenylation specific factor 3 (CPSF3), an mRNA processor [139,141]. CPSF3 forms part of the pre-mRNA cleavage and polyadenylation complex that cleaves the pre-mRNA before coupling the poly (A) tail [142,143]. CPSF3 is also involved in the histone cleavage complex that cleaves and processes pre-mRNAs of core histones without any adenylation taking place [142]. CPSF3 is also targeted by AN11736, another benzoxaborole, which is being developed as an anti-trypanosomal for the treatment of AAT caused by *T. vivax* and *T. congolense* [144]. The molecular docking of acoziborole to *T. brucei* CPSF3 reveals 26 highly conserved amino acids in close proximity to acoziborole [139]. Between human and *T. brucei*, there are only four amino acid residue substitutions at the sites that are in close proximity to docked acoziborole [139]. Therefore, the scientific explanation for the selectivity of acoziborole could be that the substitutions could decrease the drug’s affinity for the human orthologue of CPSF3.

Pafuramidine (DB289), a prodrug that is metabolized to form the active anti-trypanosomal diamidine analogue of pentamidine, furamidine (DB75), had also reached clinical trials [5,145,146]. Pafuramidine was the first orally administered drug to reach phase III clinical trials for the treatment of hemolymphatic HAT [146]. DB75, like pentamidine, is a diamidine that selectively accumulates in trypanosomes, with the DNA-containing organelles taking precedence [147]. The drug with an unknown mechanism of action was discontinued due to safety and toxicity concerns [146].

Drawing from pafuramidine, a series of other diamidine molecules have been investigated due to their anti-trypanosomal properties [5]. DB829 and DB820, which are the active metabolites derived from the prodrugs DB868 and DB844, respectively, have been determined to be efficacious against stage II HAT in animal models [148,149]. DB820 has also been determined to possess a similar mechanism of accumulation as pentamidine, accumulating in DNA-containing organelles and binding to A-T-rich sites on DNA [148].

DB829 is BBB penetrant, and an effective trypanocide within the CNS [149]. Due to its structural similarity to DB75, the development of DB829 was discontinued, but the structural differences may have rendered DB829 less toxic [150]. Since pentamidine and the other diamidines are ineffective against meningoencephalitic HAT, the efficacy of DB829 could be due to BBB transporters [150,151]. DB829 is either more effectively imported into the CNS by the transporters or the transporters that extrude pentamidine from the CNS are less efficacious against DB829 [80,150].

Due to the toxicity of melarsoprol and the inconvenience associated with its administration, melarsoprol–cyclodextrin complexes were developed as potential HAT chemotherapeutics [150,152]. The complexes, melarsoprol–hydroxypropyl-β-cyclodextrin and melarsoprol-randomly methylated-β-cyclodextrin, were determined to be efficacious against meningoencephalitic *R*-HAT when orally administered in mice [152]. This was viewed as a promising outcome and was given orphan drug status by the Food and Drug Administration (FDA) in the U.S., as well as the European Medicines Agency (EMA) [150]. Protocols for a phase II clinical trial of the melarsoprol–cyclodextrin complexes were developed; however, they never went ahead due to a lack of funding [150]. Successfully consolidating funds for studying arsenicals in humans is highly unlikely [150]. Cyclodextrins are known to improve the solubility of drugs without altering their physical properties [153].

### 6.5. Validated and Potential Drug Targets

A significant number of compounds exhibiting diverse molecular structures have shown antiparasitic effectiveness in the laboratory and are interesting lead compounds for the development of new drugs. The major validated drug targets in *T. brucei* are discussed, including other potential drug targets.

In the studies carried out on the various potential HAT drugs and their validated cellular targets, different subspecies and strains have been used, including the animal-infective *T. b. brucei*. Though phenotypically identical with *T. b. rhodesiense*, *T. b. gambiense* has been determined to be genetically divergent [154]. *The T. b. brucei* TREU927 reference strain exhibits all the known phenotypes of *T. brucei* organisms [155]. Therefore, discoveries made in studies where *T. b. brucei* is employed could be inferred for *T. b. rhodesiense.*

#### 6.5.1. N-Myristoyltransferase

A selective anti-trypanosomal drug target with orthologues in humans is N-myristoyltranferase (NMT). NMT uses myristoyl-coenzyme A (CoA) as a substrate and functions in a co- or post-translational modification process referred to as myristoylation, whereby a myristate from myristoyl-CoA is transferred to an N-terminal glycine residue [156,157,158,159]. Myristoylation promotes the interaction of proteins with membranes [157]. The knockdown of NMT arrests parasitic growth and adversely affects infectivity [160,161]. Though ubiquitous and essential in eukaryotic life, it has been established that NMT inhibition in trypanosomes is highly deleterious. The difference between NMTs of humans and *T. brucei* is that humans have two genes encoding the protein, human NMT1 and NMT2, sharing 55% and 69% similarity with *T. brucei* NMT, respectively [162,163]. In *T. brucei*, phosphatases, calpain-like proteins, the ADP-ribosylation factor (ARF), and ARF-like families of GTPases are subject to myristoylation by NMT [164,165,166]. In *L. major*, hydrophilic acylated surface proteins undergo NMT-facilitated myristoylation, together with the flagellar calcium-binding protein (FCaBP) in *T. cruzi* [167,168]. Approximately 60 proteins are subject to N-myristoylation in kinetoplastids [169].

NMT inhibition is trypanocidal in vitro and in rodent models of *T. brucei* infections [170]. Classes of NMT inhibitors, referred to as sulfonamides, represented by DDD85646 have been investigated as potential anti-parasitic drugs [163]. DDD85646 was effective in treating mice infected with *T. b. brucei* S427 and fully curative in mice infected with *T. b. rhodesiense* STIB900 [170]. However, due to the conservation of NMTs between protozoan parasites and humans, there is a challenge of non-specificity or non-selectivity by the compounds as they are also potent against human NMTs [171]. Moreover, the BBB is impermeable for DDD85646, making the drug ineffective for meningoencephalitic HAT [163]. Therefore, optimizing candidate drugs to enable CNS penetration is also of interest, with a lead study having been conducted based on DDD85646 [172]. The DDD85646 optimization derivatives were demonstrated to potentially cross the BBB, possessing anti-trypanosomal properties [172]. The DDD85646 predicted binding site in *T. brucei* NMT, according to molecular docking, it comprises 31 amino acid residues, and human NMT1 and NMT2 share 83% and 90% similarity, respectively [163].

#### 6.5.2. Ubiquitination/Proteasome

The trypanosomal protein ubiquitination and proteasomal degradation processes are also viewed as potential targets for HAT drugs [173,174]. Ubiquitination is a post-translational modification process that targets terminally damaged or aged proteins for degradation in the proteasome. Ubiquitination primarily targets lysine residues on the substrate protein, modifying the amino acid by attaching ubiquitin [175]. Ubiquitination is either mono or poly, whereby mono-ubiquitination refers to the ubiquitination of the substrate protein, and poly-ubiquitination entails the formation of ubiquitin chains whereby the lysine residues on ubiquitin are also ubiquitinated [176,177,178]. Ubiquitination is carried out by three classes of enzymes, E1 (ubiquitin-activating enzyme), E2 (ubiquitin-conjugating enzyme), and E3 (ubiquitin ligase), which act on the substrate protein consecutively, cascading from E1 down to E3 [175,179,180]. Polyubiquitination is generally accepted as a prerequisite for proteasomal degradation, and it is carried out by E2 and E3 ligases [181,182]. Mammalian cells only possess a single E1 (also referred to as ubiquitin-activating enzyme 1; UBA1), while kinetoplastids possess two enzymes. The *T. brucei* E1 proteins, TbUBA1a and TbUBA1b, share 36% and 24% sequence identity with the human UBA1, respectively [173]. TbUBA1b knockdown is highly deleterious, and it results in an arrest of ubiquitination [183,184]. There are estimated to be fifteen E2 and sixty E3 ubiquitin ligase enzymes in *T. brucei* [174,185].

In the protein ubiquitination–proteasome pathway in *T. brucei*, proteins that fail to be imported into the mitochondrion are targeted for proteasomal degradation [186]. Abrogating the ATOM69 component of the atypical mitochondrial outer membrane translocase (ATOM) complex results in the channeling of three proteins to the mitochondrion, all of which are trypanosomatid-specific and involved in the ubiquitination–proteasomal pathway [186]. The proteins are *T. brucei* ubiquitin-like domain (TbUbL1), *T. brucei* C-terminal HECT E3 ubiquitin ligase domain (TbE3HECT1), and a hypothetical protein [186]. TbE3HECT1 is an E3 ubiquitin ligase [186]. These three proteins are involved in the channeling of ATOM69-specific mitochondrial substrate proteins for degradation, preventing an accumulation of the mitochondrial proteins in the cytosol [186]. This trypanosomal mechanism places further importance on targeting the ubiquitination–proteasome pathway in trypanosomes.

Targeting the ubiquitination process at the E1 level is preferred as it is at the initiation of the cascade [187]. The trypanosomal proteasome is currently the subject of research into two orally administered azabenzoxazole compounds: GNF6702, which is an improved derivative of GNF5343, and GSK3494254 (or DDD01305143) [188,189]. The compounds have been shown to be efficacious against the viability of these three kinetoplastid pathogens, namely, *Trypanosoma brucei*, *Trypanosoma cruzi*, and *Leishmania major* (TriTryps). GNF6702 and GSK3494245 bind between the β4 and β5 subunits of the proteasome, inhibiting the chymotrypsin-like activity of subunit β5 [188,189]. There are solubility concerns with regard to orally administered GNF6702, which led to the molecule being coupled with a pyridine to yield the more soluble LXE408 [190]. The resultant LXE408 compound is also efficacious against *Leishmania tarentolae* [190]. The phase II clinical trials of LXE408 are scheduled to commence, having passed the phase I clinical trials for the treatment of PKDL [191]. GSK3494245 has also reached phase I clinical trials for visceral leishmaniasis [189].

#### 6.5.3. Cyclic Adenosine Monophosphate-Specific Phosphodiesterases B1 and B2

In *T. brucei*, the B family of phosphodiesterases (PDEB) primarily functions in the hydrolysis of cellular cyclic adenosine monophosphate (cAMP) as a negative feedback mechanism, being essential for the BSF of the parasite [192]. The PDEB family consists of two paralogs (PDEB1 and PDEB2) that share 75% similarity and are both located at the flagellar membrane [192,193]. However, PDEB2 predominantly localizes in the cytosol [192]. The simultaneous knockdown of PDEB1 and PDEB2 increased intracellular cAMP levels, which was lethal for trypanosomes [192]. Morphological defects associated with PDEB1 and PDEB2 knockdown and inhibition are multinucleation and multiflagellation [194]. This has led to the phosphodiesterases being identified as drug targets in trypanosomes. Moreover, human phosphodiesterase inhibitors are already on the market for the treatment of ailments, such as chronic obstructive pulmonary disease, cardiovascular diseases and inflammation [195]. Phosphodiesterases are highly conserved between humans and the African trypanosome; therefore, the expertise in treating human diseases serves as a foundation for targeting these enzymes as drug targets from the parasite [194]. A known inhibitor of the *T. b. brucei* PDEBs is CpdA (now referred to as NPD-001), a potent tetrahydrophthalazinone compound that results in inhibition in the nanomolar concentration range [194]. Given that the PDEBs localize in the flagellar membrane, NPD-001 is lipophilic, which potentially eliminates resistance and uptake issues [196]. NPD-001 results in an increase in intracellular cAMP, leading to parasite mortality within 3 days [196]. The sensitivity of *T. b. brucei* to NDP-001 was linked to the cAMP response proteins (CARP), of which four paralogs were implicated (CARP 1–4) [196]. A further seven CARP genes (CARP 5–11) were also identified as inducing resistance to NPD-001 in *T. b. brucei* [197]. CARP 1 and CARP 11 have an affinity to NPD-001 [197]. CARP genes are more pronounced in kinetoplastids, with CARP 3 and CARP 11 being restricted to the *Trypanosoma* genus [197,198,199]. NPD-001 is non-specific, also being inhibitory toward human PDEB4. PDEB4 inhibition leads to the suppression of the tumor necrosis factor α (TNF- α) cytokine in humans [200,201]. As such, the development of the compound as a potential anti-trypanosomal drug in humans is unlikely.

A recent study has identified the anti-trypanosomal activity of phenylpyridazinone analogs of NPD-001 that inhibit *T. brucei* PDEB1 [202]. Furthermore, a selectively inhibitory alkynamide phthalazinone molecule has also been shown to be trypanocidal, having no toxic effects toward human MRC-5 cells [203]. Despite PDEB1 being a promising drug target, there appear to be no studies to determine the effects of PDEB1 inhibitors in animal models of *T. brucei* infection.

#### 6.5.4. Oxidative Stress/Polyamine Synthesis—Trypanothione System

Trypanothione reductase (TyrR) is analogous to mammalian glutathione reductase, but these proteins share less than 50% sequence similarity [204,205]. TyrR is a flavoprotein that facilitates the reduction of trypanothione disulphide to produce a dithiol, which is functionally equivalent to glutathione in mammals and functions as an antioxidant [204,206,207]. Due to the considerable structural variation between TyrR and glutathione reductase, this trypanosomatid system presents a promising target in terms of drug discovery [205,208]. However, a major challenge with developing TyrR inhibitors is the enzyme’s enlarged hydrophobic binding site, which is not compact enough for high-affinity interactions with small-molecule inhibitors [209,210]. Also, to achieve an adequate trypanocidal effect, TyrR activity must be inhibited by up to 90% due to the enzyme’s elevated efficacy in terms of turnover [207,211]. A spiro derivative molecule has been determined to differentially inhibit trypanothione reductase, having no modulatory effect on human glutathione reductase [212]. Spiro molecules serve as a scaffold for BBB-penetrant molecules [212].

The trypanothione molecule itself is synthesized in a process facilitated by trypanothione synthetase (TyrS), which is also a validated drug target that has been reported to be essential for trypanosome survival [213,214,215]. TyrS forms part of the machinery that attaches two glutathione molecules to the polyamine spermidine to result in the formation of trypanothione [213,214]. Engineered strains of *T. cruzi* expressing elevated amounts of TyrS exhibit increased growth rates and tolerance toward oxidants and heavy metals [216]. Moreover, the strain also exhibited resistance toward Chagas drugs, beznidazole and nifurtimox [216]. Recent research has identified Ebselen as a tightly binding irreversible inhibitor of TyrS in *T. brucei*, having a trypanocidal effect on cultured *T. b. brucei* cells [217].

With regard to trypanothione production, DFMO may also hamper the molecule’s biogenesis [218]. Therefore, a TyrR inhibitor could be used in combination with DFMO in treating HAT, as with nifurtimox in the NECT [218,219]. This is probably due to DFMO inhibiting the polyamine synthesis pathway, which leads to the synthesis of spermidine, a precursor for trypanothione synthesis [218,220]. The decarboxylation of L-ornithine by ODC is rate-limiting, and it is the initial step in the synthesis of the polyamines putrescine, spermidine, and spermine [221]. Using putrescine as a substrate, spermidine synthase forms spermidine, which in turn serves as the precursor for the synthesis of spermine by spermine synthase [220]. Spermidine synthase is also validated as a drug target as it is essential for parasite survival [222]. The activities of spermidine and spermine synthases rely on the donation of the aminopropyl group by decarboxylated S-adenosylmethionine to their precursor substrates, putrescine and spermidine, respectively [220,222,223]. The decarboxylated S-adenosyl methionine is synthesized by S-adenosylmethionine decarboxylase (AdoMetDC), which is a validated drug target [221]. The activity of AdoMetDc is also rate-limiting in the polyamine synthesis pathway [221].

Therefore, the polyamine synthesis pathway in which ODC is involved could further be perturbed by inhibiting AdoMetDC, which acts downstream of ODC. A study has shown the potential of inhibiting AdoMetDC, whereby thirteen classes of compounds were identified [224]. Of the thirteen, eight compounds were shown to differentially inhibit AdoMetDC in *T. brucei* compared to the human orthologue, with some of the selective compounds potentially being CNS penetrative [224]. In mammals, AdoMetDC is an inactive proenzyme that is activated by serinolysis, which leads to the formation of β- and α-chains from the N- and C-terminal of the protein, respectively [225,226,227]. The β- and α-chains form protomers that oligomerize to form homodimers [227]. The attractiveness of inhibiting AdoMetDC arises from the fact that the enzyme is divergent in *T. brucei* compared to other eukaryotic organisms [228]. The AdoMetDC gene is duplicated in *T. brucei* to form AdoMetDC and an inactive (“dead”) peptide that is referred to as prozyme, whereby the duplicate gene products form a heterodimer [228]. The prozyme is an allosteric regulator, stimulating the activity of AdoMetDC by over a thousand-fold [228]. The prozyme interacts with a region of the N-terminus of the *T. brucei* AdoMetDC that is lacking in other eukaryotic organisms [229].

#### 6.5.5. RNA-Editing Ligase and Pteridine Ligase

RNA-editing ligase (REL1) and pteridine reductase 1 (PTR1), which are unique to trypanosomatids, have also been validated as drug targets. REL1 in *T. brucei* is essential for survival in the tsetse vector and the mammalian host [230,231]. The ligase is one of two enzymes that function downstream of one of the trypanosomatid RNA-editing mechanisms, joining together the maxi-circle transcript (mRNA) molecule after the insertion of the deletion of uridine [230,231]. A study has also been conducted whereby potential REL1 lead inhibitors were identified [232]. PTR1 plays a secondary role in the processing of folates in trypanosomatids, with dihydrofolate reductase (DHFR) and thymidylate synthase (TS) playing a primary role [233,234]. The knockdown of PTR1 leads to major morphological deficiencies and is trypanocidal [234]. PTR1 knockdown trypanosomes also exhibit reduced pathogenicity [234]. DHFR-TS inhibitors are thought to be viable drug targets for the treatment of HAT; however, the inhibition of DHFR-TS results in the upregulation of PTR1, which compensates for the role of DHFR-TS [235]. Therefore, a chemotherapeutic strategy against both PTR1 and DHFR-TS could result in a potent anti-trypanosomal treatment [236,237]. Recent studies have also identified novel trypanocidal PTR1 inhibitors when combined with DHFR inhibitors. In combination with the DHFR inhibitor WR99210, a series of molecular docking predicted inhibitors of PTR1 were shown to inhibit the growth of *T. b. brucei* Lister 427 [238]. Furthermore, the inhibitors were also potential DHFR inhibitors [238]. In a separate study, cycloguanil, which is a DHFR inhibitor, has been determined to inhibit PTR1. Derivatives of cycloguanil were also demonstrated to possess EC_50_ values in the nanomolar concentration range against *T. b. brucei* Lister 427 bloodstream forms [239]. Though PTR1 has been reported to be essential for virulence in mice, no studies have been carried out to determine the effects of PTR1 and DHFR-TS inhibitors in animal models of trypanosomal infection [234].

#### 6.5.6. Repurposing Anti-Cancer Drugs as Anti-Parasitic Treatments

The repositioning of anti-cancer agents as HAT drugs is premised upon the fact that trypanosomes rapidly proliferate within the mammalian host in a manner similar to cancer cells. Therefore, compounds that inhibit cell replication, such as DNA replication inhibitors, may have some anti-trypanosomal effects [86]. Proof of this concept is provided by pentamidine, which is an existing HAT drug that inhibits trypanosomal DNA replication [2]. Interestingly, recent research has confirmed the anti-cancer capabilities of pentamidine, even though the mechanism of action in this regard may not include DNA replication inhibition [240,241,242].

A number of glycolysis inhibitors with anti-cancer potential are active against *T. brucei* [243]. Protozoan parasites, like many cancer cells, have an upregulated glycolytic process [243]. Dichloroacetic acid (DCA), 3-bromopyruvic acid (3BP), lonidamine (LND), metformin (MET), and sirolimus (SIR) have all been determined to reduce the survival of *T. brucei* parasites [243]. Recently, a potent, novel anti-cancer agent MitoTam, a drug molecule resulting from the conjugation of a tamoxifen derivative to a triphenylphosphonium vector (TPP+), has been an efficacious trypanocidal against *T. b. brucei* at nanomolar concentrations [244,245]. MitoTam acts by disrupting the mitochondrial integrity of trypanosomes and has been determined to be minimally toxic against healthy mammalian cells [245]. MitoTam treatment was also shown to prolong the lifespan of *T. brucei*-infected mice, which when left untreated, die within eight days [245].

#### 6.5.7. Heat Shock Protein 70 and 90

The *T. brucei* molecular chaperone network is implicated in survival, differentiation, and pathogenicity, as well as coping with environmental stressors [246,247,248,249]. An analysis of the molecular chaperone machinery in *T. brucei* revealed an expansion in the number of J-proteins and Hsp70 proteins, indicating that these protein families may play a critical role in the biology of the parasite [250]. Co-chaperones of *T. brucei* Hsp90 were recently identified, and it is possible that chaperone/co-chaperone interactions could be pursued as potential drug targets [251]. Molecular chaperones, in particular the Hsp70 and Hsp90 families, have been assessed as potential anti-trypanosomal drug targets [252,253,254].

##### *Trypanosoma brucei* Hsp70

The *T. brucei* cytosolic Hsp70 complement consists of three paralogues, namely TbHsp70 and TbHsp70.c, which are heat inducible, and TbHsp70.4, which is non-inducible [246,253,254]. TbHsp70 and TbHsp70.4 were susceptible to malonganenone and nuttingin compounds, which modulated their substrate-binding capabilities, and were trypanocidal [255]. As Hsp70s are ATP-dependent, they rely on type-I or type-II J-proteins to stimulate their ATPase activity. In this regard, the compounds inhibited the TbHsp70 and TbHsp70.4 ATPase activity stimulated by the cytosolic *T. brucei* type I J-protein, Tbj2 [255]. However, this inhibition was reduced in the J-protein-stimulated ATPase activity of the constitutively expressed human cytosolic Hsp70 [255].

Methylene blue and quercetin, which are trypanocidal, were reported to inhibit the activity of TbHsp70.c [256]. TbHsp70.c is an atypical Hsp70, possessing orthologues only within the order Trypanosomatida, making it a possible drug target [256]. TbHsp70.c possesses a divergent linker region, and substitutions at otherwise highly conserved amino acid residues within the substrate-binding pocket [256]. Furthermore, TbHsp70.c lacks the EEVD motif, which is essential for interacting with TPR domain-containing co-chaperones, such as the stress-inducible phosphoprotein 1 (STI1) [250,254]. STI1 serves as a regulator of substrate channeling between Hsp70 and Hsp90, and TbHsp70.c does not interact with *T. brucei* STI1 [256,257]. The benefit of targeting Hsp70s is that they possess deep nucleotide-binding clefts and substrate-binding pockets that can be targeted with small molecules [258]. The uniqueness of TbHsp70.c in this regard could serve as a basis for the development of drugs that differentially kill trypanosomal cells.

##### *Trypanosoma brucei* Hsp90

Hsp90 isoforms are important for the survival of cancer cells, and numerous anti-Hsp90 compounds have been identified and are in the pipeline for development as anti-cancer therapies. Indeed, the Hsp90α/β inhibitor, Pimitespib, received its first approval in Japan for the treatment of progressive gastrointestinal stromal tumor (GIST) [259,260,261,262]. Some of these compounds also possess potent anti-trypanosomal properties, showing an increased binding affinity for *T. brucei* Hsp90 compared to human Hsp90 [263]. The inhibition of cytosolic *T. brucei* Hsp90, TbHsp83, is anti-trypanosomal, whereby TbHsp83 is more sensitive to Hsp90 inhibitors compared to mammalian orthologues. *T. brucei* showed about 1000-fold higher sensitivity to geldanamycin and over 50-fold higher sensitivity to the geldanamycin analog 17-AAG (17-allylamino-17-demethoxy-geldanamycin/Tanespimycin) compared to its mammalian host. Another geldanamycin analog, 17-DMAG (17-dimethylaminoethylamino- 17-demethoxygeldanamycin), cured mice of *T. brucei* infection [264,265]. TbHsp83 is essential for the survival of the parasite [249,264]. The Hsp90 inhibitors, geldamycin and radicicol, had anti-trypanosomal EC_50_ that were on par with the existing HAT drugs, DFMO, suramin, melarsoprol, and pentamidine [265]. *T. b. brucei* MiTat 1.2 strain 427 bloodstream forms were used for the study [265]. The geldanamycin derivatives, 17-AAG and 17-DMAG, also selectively inhibited trypanosomal growth at nanomolar concentrations [265]. Mammalian cells were tolerant to 17-DMAG at bloodstream concentrations as high as 2680 nM [266]. The 17-AAG derivative inhibited mitosis and cytokinesis and sensitized trypanosomes to heat stress [265]. Orally administered 17-DMAG, on the other hand, due to its optimized solubility, cured trypanosome-infected mice models within 5 days [265]. The affinity of geldanamycin and its derivatives, 17-AAG and 17-DMAG, for TbHsp83 was higher than the affinity of these compounds toward the human Hsp90 cytosolic paralogues [263]. The treatment of trypanosomes with 17-AAG phenocopies the knockdown of TbHsp83, having a negative effect on cytokinesis and kinetoplast segregation [249]. Knocking down the mitochondrial *T. brucei* Hsp90, TbHsp84, results in a loss of the kDNA [249]. Protein phosphatase 5 (PP5) influenced the susceptibility of TbHsp83 and trypanosomes to geldanamycin [264]. PP5, which is also a cytosolic protein, co-localizes with TbHsp83 under stress conditions [264]. The levels of PP5 expression were inversely proportional to the trypanocidal effects of geldanamycin [264]. The knockdown of PP5, particularly in BSFs, is highly detrimental, resulting in an 8-fold decrease in parasitic growth [267].

#### 6.5.8. Adenosine Analogues

*T. brucei* and other trypanosomatid organisms are incapable of de novo purine biosynthesis and need to salvage them from the host [268,269]. An understanding of the purine salvage machinery enables the development of drug targets [270]. *T. brucei* has a transport system that imports purines from the host [271]; these transporters include the P2 and P1 adenosine transporter families [272,273]. These transporters could facilitate the delivery of trypanocidal adenosine analogues into the cell [270].

The single P2 transporter, encoded by *TbAT1,* is responsible for melarsoprol–pentamidine cross resistance and is less desirable to target for adenosine analogue delivery [95,270]. The P1 transporter family, consisting of four members, is redundant in specificity, and trypanocidal adenosine nucleosides that preferentially traverse through them are unlikely to face resistance challenges [270]. Another aspect of these analogues’ attractiveness is that they are likely to penetrate the BBB as it possesses purine transporters [274].

Adenosine analogues such as cordycepin and tubercidin have been identified as trypanocides [275,276]. However, tubercidin is toxic to mammalian cells [276,277]. A hybrid molecule between cordycepin and tubercidin has brought about the 3′-deoxy-7-deazaadenosine analogues, which are effective against trypanosomes in mouse models, with 3-deoxytubercidin being identified as the most promising of the 3′-deoxy-7-deazaadenosine analogue derivatives [269,277]. Derivatives of 3′-deoxy-7-deazaadenosine, referred to as C6-0-alkylated 7-deazainosine analogues, have been identified as trypanocides, with potential to be developed further along the pipeline of trypanosomiasis drugs [278]. Cordycepin, when administered in conjunction with deoxycoformycin or coformycin, has been reported to be curative in mice infected with *T. b. brucei*, even when the parasites have reached the brain parenchyma [279].

## 7. Vaccine Development

As a consequence of antigenic variation, successful vaccine development for HAT is deemed unlikely [4]. This is due to the constant switching of surface antigens that could potentially be targets of the vaccine-primed immune response. VSGs also shield potential vaccine antigenic targets such as the antigenic invariant surface glycoproteins (ISGs) from the immune system’s components [280,281,282,283,284,285]. ISGs are transmembrane proteins that intercalate between the VSGs [281]. Additionally, B cell impairment and depletion by trypanosomes means that developing vaccine antigens based on memory B cells could be unlikely [286]. Examples of ISGs are ISG65 and ISG75, which present approximately 70,000 and 50,000 copies of themselves, respectively, on the trypanosome’s surface [183]. Both ISG65 and ISG75 have been studied as potential antigenic targets in *T. brucei* vaccine development, with ISG75 having been determined to induce insufficient protection [280,281,287]. Interestingly, ISG75 is a target of suramin, implicated in the endocytosis of the drug [282]. In a recent study, the immunogenic potential of ISG75 alongside *T. brucei* enolase (TbENO) was reassessed. TbENO, like with enolase in *T. cruzi* and *Leishmania*, forms part of the secretome and is highly antigenic [288]. Though proving to be highly immunogenic, both ISG75 and TbENO were ultimately determined to be inadequate as antigenic targets.

## 8. Conclusions

Sleeping sickness is a neglected tropical disease that has remained a global disease burden targeted for elimination as a public health concern by 2030 [289]. However, the interruption in both active and passive screening, especially in countries like the DRC, which currently present the highest number of cases, is likely to affect the predicted elimination time according to prediction models [289,290]. Despite positive strides made by the WHO, non-governmental agencies, and researchers, the COVID-19 pandemic created a major setback. Even with these setbacks, the Ugandan Ministry of Health reported in October 2022 that with individual case management, intense surveillance, and vector control, they had eliminated sleeping sickness caused by *T. gambiense* as a public health problem [291].

Although the number of reported cases of HAT has drastically reduced over the past few decades, the reports may not be exactly due to poor surveillance in affected areas [33]. Such areas, accounting for “invisible” cases as well as the reemergence of the disease that has occurred in the past [292], call for relentless efforts in the search for updated screening and therapeutics. Improved diagnostic tools are needed to support treatment for tests for a cure in clinical trials and for surveillance of populations in control programs [293]. The most recent drug approved for HAT treatment, fexinidazole, which is administered orally and is efficacious in both stages of the disease, is evidence of sustained efforts to find alternative treatments [129].

This study has summarized most of the promising therapeutics currently being explored and their challenges. The ideal anti-parasitic drug target would be an essential protein for parasite survival significantly different from its orthologue in the host [294]. However, the presence of an orthologue in mammals does not necessarily mean that a particular target should not be pursued [117]. Both CPSF3 and NMT are present in eukaryotic cells and are promising targets for prospective HAT drugs as well as the treatment of cancer [139,170,295,296]. The inhibition of *T. brucei* N-myristoyltransferase cured trypanosomiasis in mice, showed promising selectivity, and fulfilled most of the requirements for a novel chemotherapeutic agent for HAT [163]. TbHsp90 is similar to its human counterpart but demonstrates biochemical differences that increase inhibitor efficiency.

Another approach involves the use of host-directed strategies, which may include targeting the immune system [297]. In *T. brucei*, this would be favorable as sustained activation of the immune system may be damaging [117]. In cutaneous leishmaniasis, an immune stimulator referred to as CpG oligonucleotide D35 (CpG ODN D35) has been shown to enhance the response to treatment with the pentavalent antimonial stibogluconate in macaques infected with *L. major* [298]. The animals presented with relatively smaller lesions that healed relatively quicker when treated with CpG ODN D35 and pentavalent stibogluconate [299]. This serves as an example of how host-targeted approaches could be applied in treating diseases caused by protozoan parasites, particularly kinetoplastids.

It should also be noted that the DND*i* has also adopted an artificial intelligence (AI)-driven drug design approach in the quest to discover more therapeutics for neglected diseases. This comes at a time when AI is being employed as one of the tools by which drugs may be designed, developed, and brought to market [299]. Given the status of HAT as an NTD, the potential of AI drug design to lower costs and lessen the time for drug discovery could contribute positively to the development of efficacious HAT treatments [299].

A greater understanding of drug resistance pathways and how they could be circumvented may provide solutions to address resistance to existing drugs. A preemptory research effort into drug resistance mechanisms is needed to identify drug resistance phenotypes as they emerge. Knowledge of persister forms, dormancy, and tissue distribution dynamics is required for the development of more effective chemotherapeutic strategies and to ensure that the drugs can access all sites of infection. With the consistent and concerted efforts of the WHO as well as other public and private partnerships, the eradication of HAT as a public health concern may be a reality. If both COVID-19 and sleeping sickness are viewed as endemic in Africa, then both need to be managed concurrently.

## Figures and Tables

**Figure 1 ijms-24-12529-f001:**
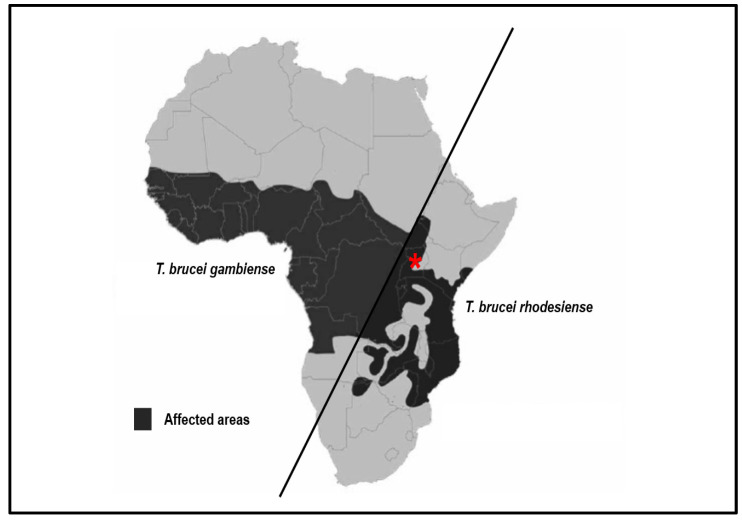
Distribution map for human African trypanosomiasis. The map illustrates the distribution of *R*-HAT and *G*-HAT (black). The bold line is a demarcation separating the regions in which *T. b. rhodesiense* and *T. b. gambiense* occur. The red asterisk serves to highlight Uganda, the only country in which both subspecies are found [13].

**Figure 2 ijms-24-12529-f002:**
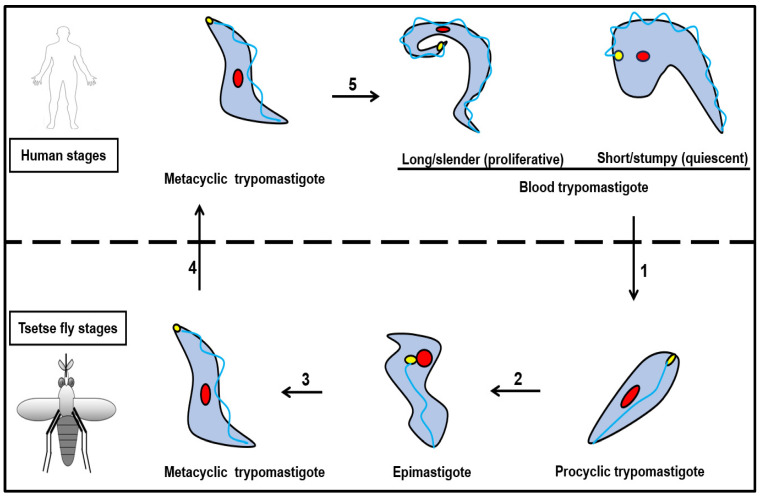
Transmission and lifecycle of the *T. brucei* parasite. The trypanosome is digenetic, shuttling between the tsetse fly and a mammalian host. (1) An uninfected tsetse fly takes up the non-proliferative short stumpy bloodstream trypomastigotes during its bloodmeal. The bloodstream trypomastigotes form procyclic trypomastigotes that multiply by binary fission. (2 and 3) The procyclic trypomastigotes transform into epimastigotes in the salivary glands where they differentiate into the human infective metacyclic trypomastigote. (4) The bite of an infected tsetse fly injects the metacyclic trypomastigote into its next mammalian host. (5) The trypomastigote is transformed into the bloodstream form, which multiplies by binary fission and spreads in the body fluids. The bloodstream forms are either present as long slender or short stumpy forms. The blue lines represent the flagellum as protruding from the flagellar pocket (shown in yellow). The nuclei of the various morphotypes are shown in red. Adapted from [4].

## Data Availability

Not applicable.

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
