# Peer review of "Tackling Sleeping Sickness: Current and Promising Therapeutics and Treatment Strategies"

_ijms, 2023, doi:10.3390/ijms241512529_

Round 1
Reviewer 1 Report
I think the review in most aspects is nicely written but there are also some parts that need to be rewritten, especially in the selection what drugs to describe (see the last point below). Here is a list of the things I found:
Introduction. There are many more species and subspecies that can cause animal african trypanosomiasis than the two subspecies mentioned (T. congolense, T. vivax, T. b. evansi etc).
Row 106. I assume it should be the 21st century (it is written 20th)
Paragraph row 124-138. The take-home message of the paragraph is unclear. The initial part indicated that rising temperatures is a problem (from a HAT point of view), but from reading the text it sounded rather lite the rising temperatures will be beneficial to prevent the spread of HAT because it will be harmful for larvae and pupae.
Fig. 2. Include the two types of bloodstream forms: long slender (proliferating) and short stumpy (non-proliferating and transmissable to the tsetse fly). Describe also what you want to show with the position of the nucleus and kinetoplast in the figure.
Row 167-168. It is not mentioned that the parasite infects the central nervous system when the text goes through all the tissues infected. I am aware about that this comes later, so it is enough to just briefly mention that this will be described later.
Row 170-171. It is not clear how the bloodstream that normally downregulate most mitochondrial activities can survive on fatty acids in adipose tissue. Another problem is that the fat is inside the cells in the adipose tissue. Mention briefly if there is any major change in life cycle stage of the parasites residing in the adipose tissue and other tissues (can they become intracellular?).
Row 224. It is written that chronic T. cruzi infections lead to heart failure etc (change to “may lead”, because otherwise it sounds like it is always the case.
Row 296. Write also that ODC is necessary for the synthesis of trypanothione, and thereby is necessary for the redox potential of the parasite.
Drug development section (row 406 and onwards). I think the selection of drug projects described is not clear. Some of the descriptions are very detailed although they have only been tested on enzymes or cells, whereas others that have been developed much further are not mentioned at all. An example of a field that has generated many promising drug candidates in recent years is nucleotide metabolism (especially adenosine analogues) and I think it would be good to include that in the description. When it comes to the already described drug development projects, clearly state at what stage they are at (clinical testing, animal experiments or testing on enzymes/cells). Drugs and drug targets only verified on enzymes/cells do not need to be described as extensively as it is now.
The English language is mostly correct, but there are also sections with awkward sentences and other mistakes so I think it could benefit from English editing. Here are some examples of errors:
Row 20: effecctive
Row 46: twenty two (it should be one word)
Row: 189: develp
Many instances with missing commas.
Author Response
Feedback-reviewer 1
I think the review in most aspects is nicely written but there are also some parts that need to be rewritten, especially in the selection what drugs to describe (see the last point below). Here is a list of the things I found:
Introduction. There are many more species and subspecies that can cause animal african trypanosomiasis than the two subspecies mentioned (T. congolense, T. vivax, T. b. evansi etc).
The first paragraph has been rewritten to state that T. brucei brucei is one of several trypanosomes that cause AAT
Row 106. I assume it should be the 21st century (it is written 20th)
This has been corrected to 21st century
Paragraph row 124-138. The take-home message of the paragraph is unclear. The initial part indicated that rising temperatures is a problem (from a HAT point of view), but from reading the text it sounded rather lite the rising temperatures will be beneficial to prevent the spread of HAT because it will be harmful for larvae and pupae.
This section is confusing and has been removed
Fig. 2. Include the two types of bloodstream forms: long slender (proliferating) and short stumpy (non-proliferating and transmissable to the tsetse fly). Describe also what you want to show with the position of the nucleus and kinetoplast in the figure.
The positioning of the nucleus and flagellar pocket have been explained below the figure. The figure has been corrected as well as the figure legend to distinguish between the long and stumpy forms.
Row 167-168. It is not mentioned that the parasite infects the central nervous system when the text goes through all the tissues infected. I am aware about that this comes later, so it is enough to just briefly mention that this will be described later.
This has been changed, the following sentence was included - At the advanced stage of the parasitic infection and pathology, the parasites are also present in the cerebrospinal fluid, having infiltrated the central nervous system (CNS).
Row 170-171. It is not clear how the bloodstream that normally downregulate most mitochondrial activities can survive on fatty acids in adipose tissue. Another problem is that the fat is inside the cells in the adipose tissue. Mention briefly if there is any major change in life cycle stage of the parasites residing in the adipose tissue and other tissues (can they become intracellular?).
This comment has been addressed. The trypanosomes reside within the interstitial spaces of the adipose tissues, and they are always extracellular. The following paragraph was added to the text
Within the adipose tissues, the parasites occupy interstitial spaces, either between adjacent adipocytes, or flanked by an adipocyte and a capillary [45]. The parasites likely to scavenge for free fatty acids released by the adipocytes via mechanisms that are yet to be understood [52]. In terms of energy requirements within the adipocytes, the parasites could be reliant on fatty acid β-oxidation pathways [45]. In this regard, T. brucei has been demonstrated to possess fatty acid β-oxidation capabilities, but it has not yet been determined if this pathway serves to fulfill the energy requirements of the adipocyte tissue forms [45].
Row 224. It is written that chronic T. cruzi infections lead to heart failure etc (change to “may lead”, because otherwise it sounds like it is always the case.
This section has been deleted as reviewer 2 wanted us to reduce the size of the first half of the document
Row 296. Write also that ODC is necessary for the synthesis of trypanothione, and thereby is necessary for the redox potential of the parasite
This was included in the text
Drug development section (row 406 and onwards). I think the selection of drug projects described is not clear. Some of the descriptions are very detailed although they have only been tested on enzymes or cells, whereas others that have been developed much further are not mentioned at all. An example of a field that has generated many promising drug candidates in recent years is nucleotide metabolism (especially adenosine analogues) and I think it would be good to include that in the description. When it comes to the already described drug development projects, clearly state at what stage they are at (clinical testing, animal experiments or testing on enzymes/cells). Drugs and drug targets only verified on enzymes/cells do not need to be described as extensively as it is now.
The adenosine analogues were included as a new section. The details of the testing were included but it was not always clear from the papers which stage or cells etc were used
Comments on the Quality of English Language
The English language is mostly correct, but there are also sections with awkward sentences and other mistakes so I think it could benefit from English editing. Here are some examples of errors:
Row 20: effective
Corrected to effective
Row 46: twenty two (it should be one word)
This was removed
Row: 189: develop
This was corrected
Many instances with missing commas.
Reviewer 2 Report
Dear authors,
This manuscript is very interesting, however, there are some concerns that need to be adressed. first of all, the WHO targets need to be correctly stated, they are different for G-HAT and for R-HAT, and you need to mention systematically througout the paper if we are talking about G-HAT or R-HAT. Since you are using both in your manuscript, it can be confusing for non-expert readers. The manuscript is very long, and in the beginning it seems like this going to be a full review about HAT while towards the end, the manuscript is the description of the HAT-molecules and challenges. I propose to condense the first part of the manuscript, and focus on the second part of the manuscript. Ask yourself is it really necessary to explain the life cycle of the parasite here if the core of the manuscript is about therapeutics? therefore the title is also misleading since challenges to interrupt transmission is not only about the therapeutics, there is so much more to it, like diagnostics, screening strategies, etc. Moreover, there is many outdated information during the first part of your manuscript.
Another suggestion would be to group the information on G-HAT and group the information on R-HAT. the way it is presented now is confusing.
Thank you

Author Response
Feedback – reviewer 2
This manuscript is very interesting, however, there are some concerns that need to be adressed. first of all, the WHO targets need to be correctly stated, they are different for G-HAT and for R-HAT, and you need to mention systematically througout the paper if we are talking about G-HAT or R-HAT. Since you are using both in your manuscript, it can be confusing for non-expert readers. The manuscript is very long, and in the beginning it seems like this going to be a full review about HAT while towards the end, the manuscript is the description of the HAT-molecules and challenges. I propose to condense the first part of the manuscript, and focus on the second part of the manuscript. Ask yourself is it really necessary to explain the life cycle of the parasite here if the core of the manuscript is about therapeutics? therefore the title is also misleading since challenges to interrupt transmission is not only about the therapeutics, there is so much more to it, like diagnostics, screening strategies, etc. Moreover, there is many outdated information during the first part of your manuscript.
Another suggestion would be to group the information on G-HAT and group the information on R-HAT. the way it is presented now is confusing.
These comments are valid and the manuscript has been condensed as far as possible. A description of the lifecycle helps in understanding the development of the parasite and the section on the drugs and other targets. The title was misleading and has been revised. As far as possible we have stated whether the drug is for G-HAT or R-HAT.
Round 2
Reviewer 1 Report
I think all major concerns are addressed, but there are still some minor corrections that need to be performed:
Row 30 (One of the causative agents of the disease…). It sounds better to just say: “The causative agents of the human disease…..”, and thereby avoid that the reader wonders why the other ones are not mentioned.
Row 119. Change “fly” to “flies”.
Row 128 and many other places. The bloodstream forms are generally named “long slender bloodstream form” and “short stumpy bloodstream form” (excluding “and” in the name). I think it is a bit confusing in the text that the metacyclics also are referred to as short and stumpy.
Row 158. There is an extra space in front of “posterior”.
Row 164. It sounds awkward with “extracellular parasite T. brucei”. I propose to just write T. brucei here and instead include “extracellular” in the first sentence of the paragraph: “T. brucei is an extracellular parasite that resides and multiplies in…..”
Row 166 (and row 194). The abbreviations BSF and PCF need to be introduced the first time they are mentioned.
Row 180. Remove “and pathology”.
Row 197. It should be “the infection”.
Row 231. Change “that” to “those”.
Row 257. Correct the spelling of eflornithine.
Row 268. Add “activity” after antitrypanosomal.
Row 329. Add a hyphen: “endo- and exocytosis”.
Row 344. Correct to “an AQP2-lacking” (note that there are times where "AQP2-lacking" is mentioned that need to be corrected).
Row 398-399. CNS should also be mentioned here (or alternatively indicate that the description is for early stages of the disease).
Row 400-401. It is confusing that it is first mentioned that the adipose tissue is immune privileged and then that it triggers a broad immune response.
Row 411. Correct to “adipose” (instead of adipocyte).
Row 430. I think a word is missing after phase II. I also think the switch to Chagas disease is quite abrupt and it would be good to have an introductory sentence first.
Row 451. Correct “T. congolense” (it is a capital C in the manuscript).
Row 533. It is only E3 that is a ubiquitin ligase (E1 and E2 are other components of the ubiquitinylation system).
Row 558. By saying “the three kinetoplastid pathogens” it sounds like there are no other human kinetoplastid pathogens, but there are several human pathogenic Leishmania species.
Row 581. Rephrase “the African trypanosome” (there are several African trypanosomes).
Row 588. Are there any animal studies on CpdA? Mention in the text if animal studies have been performed or not.
Row 600. What is a “spiro containing derivative molecule”?
Row 666. Are there any animal studies with PTR1 and DHFR inhibitors? Mention in the text if animal studies have been performed or not.
Row 670. Correct “effects”.
Row 668-678. I strongly recommend to remove the entire section since there seems to be no drugs developed.
Row 679-690. Also this section should preferably be removed since there are no drugs mentioned.
Row 702. Correct to “process”.
Row 708. Are there any animal studies with the drugs mentioned? Mention in the text if animal studies have been performed or not.
Row 380. “Order” should be non-capital.
Row 718-777. The descriptions of Hsp70 and Hsp90 inhibitors are quite difficult to read and contain too much information.
Row 757. Rephrase “cured mice of T. brucei infection”.
Row 759. Correct to “The Hsp90 inhibitors”.
Row 761. Correct to “bloodstream forms” or BSFs.
Row 762. Correct to “The geldanamycin derivatives 17-AGG and 17-DMAG” (article added and comma removed).
Row 767-777. This part could perhaps be removed (it makes the message weaker).
Row 791. Rephrase to “likely to penetrate”.
Row 791. It is mentioned that the adenosine analogues are likely to penetrate BBB but in fact it has been proven that some of them also work against the CNS stage of the disease. I think this should be mentioned.
As listed above, there were still quite a lot remaining mistakes although I had mentioned already the first time that I recommended language editing.
Author Response
Round 2 corrections
Reviewer 1
Comments and Suggestions for Authors
I think all major concerns are addressed, but there are still some minor corrections that need to be performed:
Row 30 (One of the causative agents of the disease…). It sounds better to just say: “The causative agents of the human disease…..”, and thereby avoid that the reader wonders why the other ones are not mentioned.
This was changed
Row 119. Change “fly” to “flies”.
This was corrected
Row 128 and many other places. The bloodstream forms are generally named “long slender bloodstream form” and “short stumpy bloodstream form” (excluding “and” in the name). I think it is a bit confusing in the text that the metacyclics also are referred to as short and stumpy.
This was addressed as the word ‘and’ has been excluded.
Row 158. There is an extra space in front of “posterior”.
corrected
Row 164. It sounds awkward with “extracellular parasite T. brucei”. I propose to just write T. brucei here and instead include “extracellular” in the first sentence of the paragraph: “T. brucei is an extracellular parasite that resides and multiplies in…..”
This was changed
Row 166 (and row 194). The abbreviations BSF and PCF need to be introduced the first time they are mentioned.
This has been addressed.
Row 180. Remove “and pathology”.
This has been corrected.
Row 197. It should be “the infection”.
This has been corrected.
Row 231. Change “that” to “those”.
This has been corrected.
Row 257. Correct the spelling of eflornithine
This has been corrected.
Row 268. Add “activity” after antitrypanosomal.
This has been corrected.
Row 329. Add a hyphen: “endo- and exocytosis”.
This has been corrected.
Row 344. Correct to “an AQP2-lacking” (note that there are times where "AQP2-lacking" is mentioned that need to be corrected).
This has been corrected.
Row 398-399. CNS should also be mentioned here (or alternatively indicate that the description is for early stages of the disease).
Row 400-401. It is confusing that it is first mentioned that the adipose tissue is immune privileged and then that it triggers a broad immune response.
This has been addressed.
Row 411. Correct to “adipose” (instead of adipocyte).
This has been corrected.
Row 430. I think a word is missing after phase II. I also think the switch to Chagas disease is quite abrupt and it would be good to have an introductory sentence first.
This has been corrected.
Row 451. Correct “T. congolense” (it is a capital C in the manuscript).
This has been corrected.
Row 533. It is only E3 that is a ubiquitin ligase (E1 and E2 are other components of the ubiquitinylation system).
This has been corrected.
Row 558. By saying “the three kinetoplastid pathogens” it sounds like there are no other human kinetoplastid pathogens, but there are several human pathogenic Leishmania species.
This has been corrected.
Row 581. Rephrase “the African trypanosome” (there are several African trypanosomes).
This has been corrected.
Row 588. Are there any animal studies on CpdA? Mention in the text if animal studies have been performed or not.
This has been corrected.
Row 600. What is a “spiro containing derivative molecule”?
This has been addressed.
Row 666. Are there any animal studies with PTR1 and DHFR inhibitors? Mention in the text if animal studies have been performed or not.
This has been addressed.
Row 670. Correct “effects”.
This has been corrected.
Row 668-678. I strongly recommend to remove the entire section since there seems to be no drugs developed.
Section removed.
Row 679-690. Also this section should preferably be removed since there are no drugs mentioned.
Section removed.
Row 702. Correct to “process”.
This has been corrected.
Row 708. Are there any animal studies with the drugs mentioned? Mention in the text if animal studies have been performed or not.
This has been corrected.
Row 380. “Order” should be non-capital.
This has been corrected.
Row 718-777. The descriptions of Hsp70 and Hsp90 inhibitors are quite difficult to read and contain too much information.
Row 757. Rephrase “cured mice of T. brucei infection”.
This has been corrected.
Row 759. Correct to “The Hsp90 inhibitors”.
This has been corrected.
Row 761. Correct to “bloodstream forms” or BSFs
This has been corrected.
Row 762. Correct to “The geldanamycin derivatives 17-AGG and 17-DMAG” (article added and comma removed).
This has been corrected.
Row 767-777. This part could perhaps be removed (it makes the message weaker).
This has been addressed by removing the text that could weaken the message that we seek to convey.
Row 791. Rephrase to “likely to penetrate”
This has been corrected.
Row 791. It is mentioned that the adenosine analogues are likely to penetrate BBB but in fact it has been proven that some of them also work against the CNS stage of the disease. I think this should be mentioned.
This has been addressed.

Reviewer 2 Report
Dear Authors,
Unfortunately the majority of the detailed comments were not adressed.
It was also difficult to assess the changes made to the manuscript since there was no tracked-changed document.
Please revise the manuscript.
Best regards

Author Response
Reviewer 2
Comments and Suggestions for Authors
Dear Authors,
Unfortunately the majority of the detailed comments were not adressed.
It was also difficult to assess the changes made to the manuscript since there was no tracked-changed document.
Please revise the manuscript.
We wanted to differentiate this paper from others by including other factors that should be considered such as the phenomena of persistors”” and drug delivery problems. We have also included drugs that did not reach clinical trials with reasons if available.
The manuscript has been revised, whereby changes are shown in coloured font.
Line 93 – The WHO target has been specified for G-HAT, the rest of the paper also indicates if we are referring specifically to G-HAT or R-HAT
In one of our previous paper Cell Stress and Chaperones (2019) 24:125–148 https://doi.org/10.1007/s12192-018-0950-x, we found that it was not always possible to find information pertaining to R-Hat, so we included this statement below:
In the studies carried out on the various potential HAT drugs and their validated cellular targets, different subspecies and strains have been used, including the animal infective T. b. brucei. Though phenotypically identical with T. b. rhodesiense, T. b. gambiense has been determined to be genetically divergent (Jackson et al., 2010). The T. b. brucei TREU927 reference strain exhibits all the known phenotypes of T. brucei organisms (Gibson, 2012). Therefore, discoveries made in studies where T. b. brucei is employed could be inferred for T. b. rhodesiense.
We have included a similar statement in the revised manuscript (line 5040.
References
Gibson, W., 2012. The origins of the trypanosome genome strains Trypanosoma brucei brucei TREU 927, T. b. gambiense DAL 972, T. vivax Y486 and T. congolense IL3000. Parasites Vectors 5, 71. https://doi.org/10.1186/1756-3305-5-71
Jackson, A.P., Sanders, M., Berry, A., McQuillan, J., Aslett, M.A., Quail, M.A., Chukualim, B., Capewell, P., MacLeod, A., Melville, S.E., Gibson, W., Barry, J.D., Berriman, M., Hertz-Fowler, C., 2010. The Genome Sequence of Trypanosoma brucei gambiense, Causative Agent of Chronic Human African Trypanosomiasis. PLoS Negl Trop Dis 4, e658. https://doi.org/10.1371/journal.pntd.0000658
I have attached the manuscript with tracked changes.
